# Electrosynthesis of a nylon-6 precursor from cyclohexanone and nitrite under ambient conditions

Yongmeng Wu [1,4] ✉, Jinghui Zhao[1,4], Changhong Wang[2], Tieliang Li [1], Bo-Hang Zhao[1], Ziyang Song[1], Cuibo Liu [1] & Bin Zhang [1,3] ✉

Cyclohexanone oxime, an important nylon-6 precursor, is conventionally synthesized through cyclohexanone-hydroxylamine ($NH_2OH$) and cyclohexanone ammoxidation methodologies. These strategies require complicated procedures, high temperatures, noble metal catalysts, and toxic $SO_2$ or $H_2O_2$ usage. Here, we report a one-step electrochemical strategy to synthesize cyclohexanone oxime from nitrite ($NO_2^-$) and cyclohexanone under ambient conditions using a low-cost Cu-S catalyst, avoiding complex procedures, noble metal catalysts and $H_2SO_4/H_2O_2$ usage. This strategy produces 92% yield and 99% selectivity of cyclohexanone oxime, comparable to the industrial route. The reaction undergoes a $NO_2^- \rightarrow NH_2OH \rightarrow$ oxime reaction pathway. This electrocatalytic strategy is suitable for the production of other oximes, highlighting the methodology universality. The amplified electrolysis experiment and techno-economic analysis confirm its practical potential. This study opens a mild, economical, and sustainable way for the alternative production of cyclohexanone oxime.

Cyclohexanone oxime is a key precursor for caprolactam production, which is the monomer for the synthesis of nylon-6. Global production of nylon-6 is forecasted to reach 8.9 million tons per year by 2024; thus, the demand for cyclohexanone oxime will increase accordingly[1,2]. At present, more than 90% of cyclohexanone oxime in the world is produced by the traditional cyclohexanone-hydroxylamine ($NH_2OH$) method (Fig. 1a)[3]. This strategy includes two steps: (1) $NO_x$ is reduced by $H_2$ or $SO_2$ to synthesize $NH_2OH$; (2) $NH_2OH$ reacts with cyclohexanone to form cyclohexanone oxime. The former step requires explosive $H_2$ and corrosive $SO_2$ and $NO_x$, causing concerns about safety, cost, and sustainability.

Accordingly, alternative strategies have been developed (Supplementary Fig. 1)[4,5]. Cyclohexanone ammoxidation is the most promising strategy in industrial production (Fig. 1b). In this method, $NH_2OH$ is produced in situ via the oxidation of $NH_3$ by $H_2O_2$ and

subsequently reacting noncatalytically with cyclohexanone to produce cyclohexanone oxime. Excessive $H_2O_2$ is typically required due to its low stability under the associated reaction conditions (e.g., elevated temperatures, high pH), resulting in high cost and easy deactivation problems. Very recently, Lewis et al. successfully used in situ-generated $H_2O_2$ from $H_2$ and $O_2$ to replace the performed $H_2O_2$ to realize cyclohexanone oxime production (Fig. 1c)[6]. This approach eliminates the necessity of $H_2O_2$ transportation and storage but requires a noble metal catalyst, $H_2$, and an elevated temperature. Therefore, it is highly desirable to develop an alternative strategy to achieve the sustainable, mild, and efficient synthesis of cyclohexanone oxime.

Electrochemistry has emerged as an attractive strategy in synthetic chemistry[7–15]. The electrochemical reduction of nitrogen oxides ($NO, NO_3^-, NO_2^-$, etc.) into the lowest-valence-state ammonia ($NH_3$) has

[1]Department of Chemistry, Institute of Molecular Plus, School of Science, Tianjin University, Tianjin 300072, China. [2]College of Engineering, Hebei Normal University, Hebei 050024, China. [3]Tianjin Key Laboratory of Molecular Optoelectronic Science, Key Laboratory of Systems Bioengineering (Ministry of Education), Tianjin University, Tianjin 300072, China. [4]These authors contributed equally: Yongmeng Wu, Jinghui Zhao. ✉e-mail: ymwu01@tju.edu.cn; bzhang@tju.edu.cn

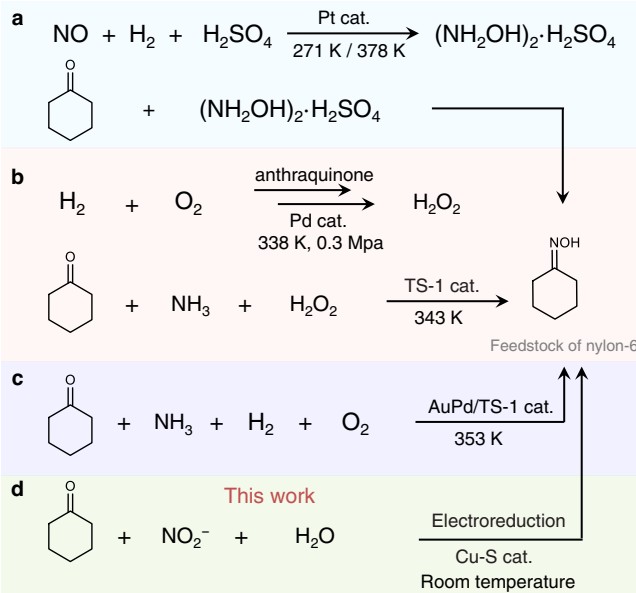

**Fig. 1 | Schematic diagrams of cyclohexanone oxime synthesis.**
**a** cyclohexanone-hydroxylamine method, (**b**) cyclohexanone ammoxidation method, (**c**) recently reported method and (**d**) the proposed electrosynthesis of cyclohexanone oxime.

made great advances[16–20]. In most studies, $NH_2OH^*$ is revealed as the intermediate during the nitrogen oxide electroreduction process, but $NH_2OH^*$ is rather unstable and easily reduced to $NH_3$. Thus, the in situ utilization of the generated $NH_2OH^*$ during nitrogen oxide electroreduction to synthesize organic compounds, especially oxime, is significant but rarely reported[21]. Inspired by recent advances in nitrogen oxide electroreduction, we speculate that utilizing the $NH_2OH^*$ formed in situ by electroreduction of nitrogen oxides to react with cyclohexanone to form cyclohexanone oxime can be a promising alternative route for the facile synthesis of cyclohexanone oxime at room temperature. Generally, using gaseous nitrogen oxides as reactants often exhibits a low single-pass conversion rate under ambient conditions; thus, it is desirable to use liquid feedstocks to achieve oxime electrosynthesis with a high conversion rate.

Herein, we report an electrochemical method to synthesize cyclohexanone oxime from nitrite and cyclohexanone over a sulfur-modified Cu (Cu-S) cathode (Fig. 1d). This reaction proceeds in water under ambient conditions and avoids complex procedures and $H_2O_2$ usage. A 92% yield and 99% selectivity (in terms of C) of cyclohexanone oxime are achieved at a potential of −0.9 V vs. Ag/AgCl. The high performance can be maintained after 50 cycles of tests, verifying the catalyst's durability. A series of control experiments, in situ attenuated total reflection Fourier transform infrared (in situ ATR-SEIRAS) spectroscopy, and density functional theory (DFT) calculations reveal a $NO_2^- \rightarrow NH_2OH \rightarrow$ oxime pathway. Furthermore, a two-electrode circular flow electrolyzer can deliver 40 mmol of cyclohexanone oxime at a constant current density of 50 mA cm$^{-2}$ within 6 h, highlighting the application potential.

## Results

### Cyclohexanone oxime electrosynthesis over a Cu-S cathode
Cu materials have been reported to be highly active in the electrochemical reduction of $NO_3^-/NO_2^-$ with $H_2O$ to ammonia ($NH_4^+$)[22]. To make the electrochemical reduction reaction stay at the $NH_2OH$ step for synthesizing cyclohexanone oxime, the reduction ability of the Cu catalyst should be weakened. Recently, we reported that the introduction of S into Cu (Cu-S) weakens the electroreduction ability of Cu and inhibits the overhydrogenation of alkenes[23]. Thus, we propose that

Cu-S is an electrocatalyst candidate for using $NO_2^-$ electroreduction to produce $NH_2OH$ in situ for cyclohexanone oxime.

The Cu-S electrocatalyst was synthesized and characterized by scanning electron microscopy (SEM) and X-ray photoelectron spectroscopy (XPS) (Supplementary Figs. 2, 3 and Supplementary Note 1). The catalytic performance was tested in an H-type cell using 20 mL of 0.5 M phosphate buffer solution (PBS) containing 0.2 mmol cyclohexanone and 2 mmol $NaNO_2$ (Supplementary Fig. 4). The linear sweep voltammetry (LSV) curve shows an enhanced current density after the addition of cyclohexanone and $NaNO_2$ (Supplementary Fig. 5). After 4000 s of electrolysis at different potentials, the products were analysed and quantified by $^1H$ nuclear magnetic resonance ($^1H$ NMR), gas chromatography–mass spectrometry (GC–MS), and gas chromatography (GC). Interestingly, cyclohexanone oxime was identified as the only organic product. All the peaks at 2.4, 2.1, 1.6, and 1.5 ppm in the $^1H$ NMR spectrum (Fig. 2a, Supplementary Fig. 6 and Supplementary Note 2) and 160.7, 32.1, 26.8, 25.7, 25.5, and 24.4 ppm in the $^{13}C$ NMR spectrum (Fig. 2b, Supplementary Fig. 7 and Supplementary Note 3) match well with the cyclohexanone oxime standard sample. Additionally, the molecular weight of 113.1 given by GC–MS further confirms the successful synthesis of cyclohexanone oxime (Fig. 2c and Supplementary Fig. 8). A 92% yield, 99% selectivity (in terms of C), 26% FE, and 0.165 mmol h$^{-1}$ cm$^{-2}$ formation rate of cyclohexanone oxime are obtained at the optimum potential of −0.9 V vs. Ag/AgCl, and $NH_4^+$ is the major byproduct (Fig. 2d, e, Supplementary Figs. 9–12 and Supplementary Note 4). The performance is better than that of the pure Cu catalyst (Supplementary Figs. 13 and 14 and Supplementary Note 5). Then, the reaction process is monitored (Fig. 2f). Cyclohexanone is consumed completely within 4000 s, and the yield of cyclohexanone oxime shows an opposite tendency compared to cyclohexanone. Then, a preliminary techno-economic analysis (TEA) on plant-gate levelized cost per tonne of cyclohexanone oxime was performed (Supplementary Note 6). The electrosynthesis strategy is much more profitable at the optimum condition of −0.9 V vs. Ag/AgCl at a given electricity price of 10 cents kWh$^{-1}$ (the price of electricity generated from renewable sources is 3 cents kWh$^{-1}$) (Fig. 2g)[24], showing the industrial application potential of this electrosynthesis strategy. Notably, 10-fold molar equivalents of $NaNO_2$ were used to realize the rapid synthesis of cyclohexanone oxime, and the cost should be further decreased by improving the utilization of $NaNO_2$ (Supplementary Fig. 15). Subsequently, the durability of the catalyst was assessed, and the performance and catalyst structure were well maintained during 50 cyclic tests, showing the good stability of the Cu-S catalyst (Fig. 2h) (Supplementary Figs. 16, 17).

### Mechanistic studies of cyclohexanone oxime electrosynthesis
The reaction mechanism was investigated. To confirm the origin of cyclohexanone oxime production, $D_2O$ and $Na^{15}NO_2$ were used as the H and N sources to replace $H_2O$ and $NaNO_2$, respectively (Entries 1, 2 in Table 1). $^1H$ NMR, $^{15}N$ NMR, and GC–MS (Fig. 3a–d) demonstrate the acquisition of deuterated and $^{15}N$-labeled cyclohexanone oxime. Meanwhile, no cyclohexanone oxime is detected when removing electricity, cyclohexanone, and $NaNO_2$. These results demonstrate that cyclohexanone oxime production is an electrically driven process with $NaNO_2$ and cyclohexanone as the N and C sources, respectively (Entries 3–5 in Table 1).

The reaction pathway was elucidated by control experiments and in situ ATR-SEIRAS. At a reaction potential of −0.9 V vs. Ag/AgCl, $NH_2^*$ (1260 cm$^{-1}$) and $NH_2OH^*$ (1199 cm$^{-1}$) were detected by in situ ATR-SEIRAS using cyclohexanone and $NaNO_2$ as the raw materials (Fig. 3e)[25,26]. Because the wide $H_2O$ peak at approximately 1650 cm$^{-1}$ overlaps with the peaks of C = N and NO, we conducted the test in $D_2O$. The vibration bands at 1690 cm$^{-1}$, 1573 cm$^{-1}$, and 1481 cm$^{-1}$, assigned to the stretching vibrations of C = N, NO, and O-H in oxime, appear (Fig. 3f). Furthermore, isotope-labeling in situ ATR-SEIRAS

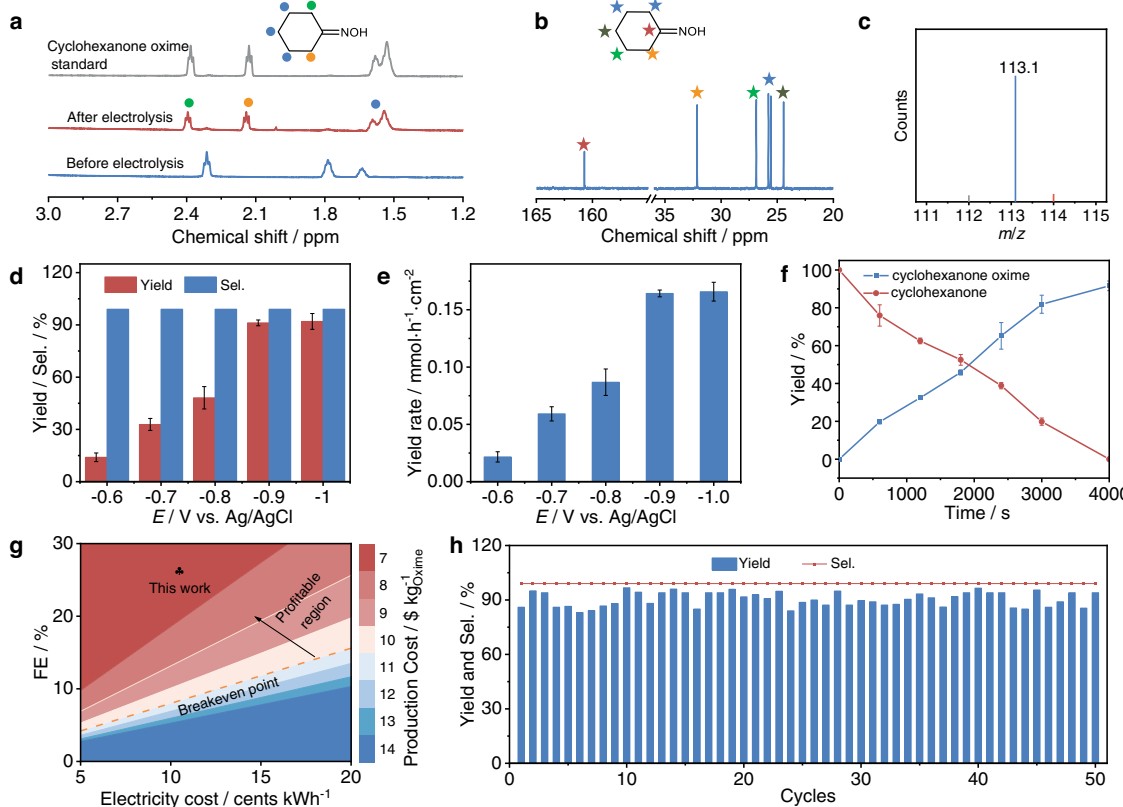

**Fig. 2 | Performance of cyclohexanone oxime electrosynthesis over a Cu-S cathode. a** $^1$H NMR, (**b**) $^{13}$C NMR, and (**c**) GC–MS detection of cyclohexanone oxime product. **d** Potential-dependent cyclohexanone oxime yield and selectivity. **e** Potential-dependent cyclohexanone oxime yield rates. **f** Time-dependent cyclohexanone conversion and cyclohexanone oxime yield. **g** Plant-gate levelized cost per tonne of cyclohexanone oxime from TEA at −0.9 V vs. Ag/AgCl. **h** Durability test at −0.9 V vs. Ag/AgCl.

## Table 1 | List of control experiments

| Entry | N-source | H-source | C-source | Electricity | Product |
|---|---|---|---|---|---|
| 1 | $NO_2^-$ | $D_2O$ | Cyclohexanone | Yes | D-cyclohexanone oxime |
| 2 | $^{15}NO_2^-$ | $H_2O$ | Cyclohexanone | Yes | $^{15}$N-cyclohexanone oxime |
| 3 | $NO_2^-$ | $H_2O$ | – | Yes | × |
| 4 | – | $H_2O$ | Cyclohexanone | Yes | × |
| 5 | $NO_2^-$ | $H_2O$ | Cyclohexanone | No | × |
| 6 | $NH_4^+$ | $H_2O$ | Cyclohexanone | Yes | × |
| 7 | NO | $H_2O$ | Cyclohexanone | Yes | √ |
| 8 | $NH_2OH$ | $H_2O$ | Cyclohexanone | Yes | √ |
| 9 | $NH_2OH$ | $H_2O$ | Cyclohexanone | No | √ |

√ indicates that cyclohexanone oxime is generated, and x indicates that no cyclohexanone oxime is detected.

experiments using Na$^{15}$NO$_2$ as the N source were conducted to verify the above analysis. The vibrations of $^{15}$NO* (1558 cm$^{-1}$), $^{15}$NH$_2$* (1232 cm$^{-1}$), $^{15}$NH$_2$OH* (1168 cm$^{-1}$), and C = $^{15}$N (1654 cm$^{-1}$) shift to lower wavenumbers by 20−40 cm$^{-1}$, while the vibration of O-H remains unchanged (Fig. 3g, h). These blueshifts are attributed to the isotope effect[27]. These results confirm the successful synthesis of cyclohexanone oxime and the formation of NO*, NH$_2$*, and NH$_2$OH* during the electroreduction process, which may serve as the active intermediate for oxime formation.

To verify the N-containing active species for cyclohexanone oxime formation, control experiments using cyclohexanone as the C source and NO, NH$_2$OH, and NH$_4^+$ as the N sources were carried out under standard conditions. No cyclohexanone oxime was detected when using NH$_4^+$ as the N source, excluding the involvement of NH$_3$ in cyclohexanone oxime formation (Entry 6 in Table 1). However, when

using NO or NH$_2$OH as the N source, both can produce cyclohexanone oxime products (Entries 7 and 8 in Table 1). Considering that NH$_2$OH is the more reduced intermediate than NO in NO$_2^-$ electroreduction, it is reasonable to regard NH$_2$OH as the active N-containing species for cyclohexanone oxime formation. Control experiments reveal that when NH$_2$OH and cyclohexanone are mixed at room temperature, cyclohexanone oxime is immediately generated even without electricity, indicating that the condensation of NH$_2$OH and cyclohexanone is a spontaneous process (Entry 9 in Table 1). This inspired us to explore whether cyclohexanone oxime can be formed by adding cyclohexanone at the end of NO$_2^-$ electroreduction. As a result, no cyclohexanone oxime was detected. We speculate that the adsorbed hydroxylamine is difficult to desorb from the catalyst surface into the electrolyte solution to react with cyclohexanone. To prove this hypothesis, we performed an electrolysis experiment of

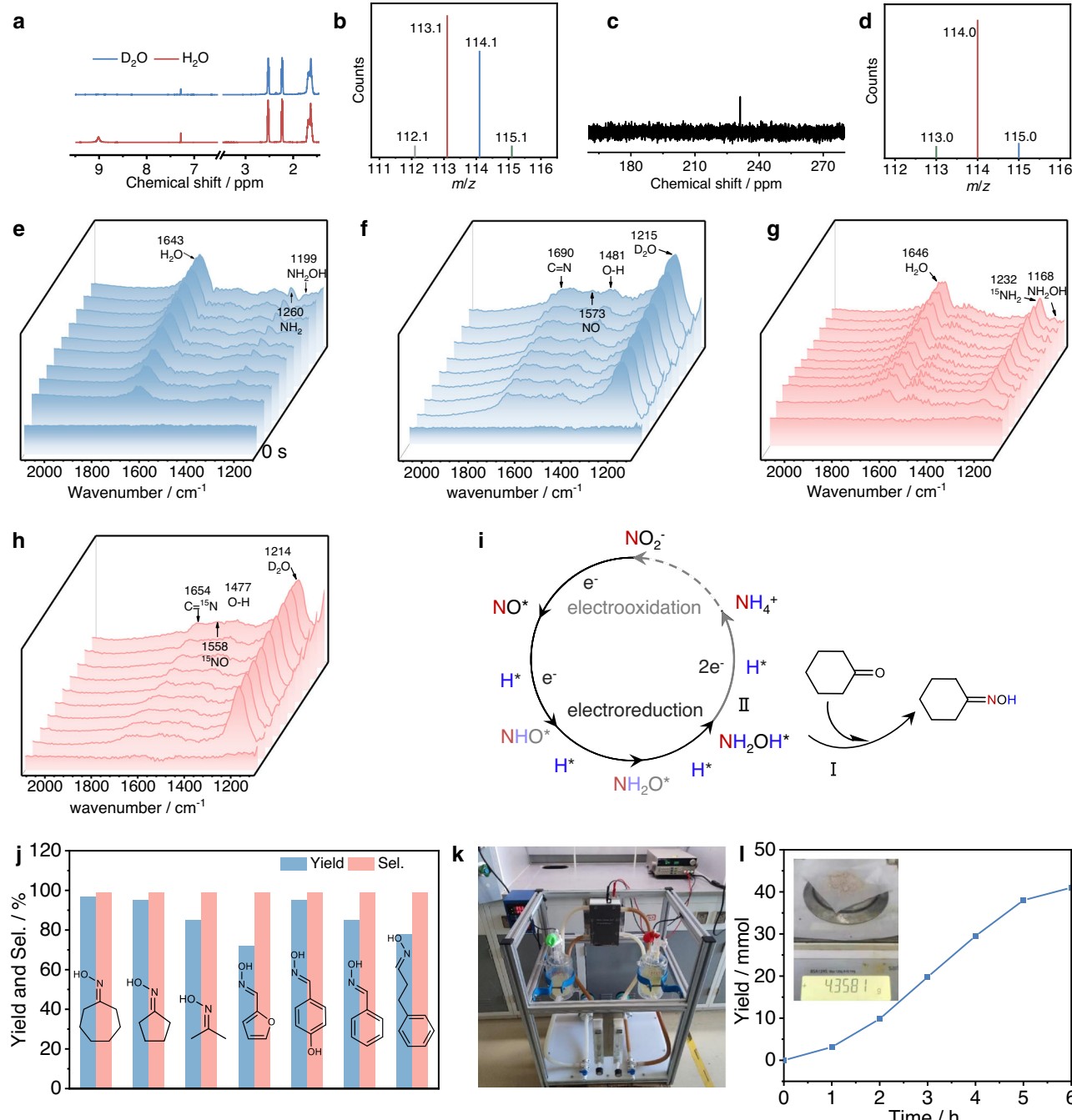

**Fig. 3 | Mechanism and application studies. a** $^1$H NMR spectra of cyclohexanone oxime (red) and deuterated cyclohexanone oxime (blue). **b** GC–MS spectrum of deuterated cyclohexanone oxime. **c** $^{15}$N NMR spectrum of $^{15}$N-cyclohexanone oxime. **d** GC–MS detection of the $^{15}$N-cyclohexanone oxime product. Time-dependent in situ ATR-SEIRAS using (**e**) $^{14}NO_2^-$ as the N-source and $H_2O$ as the H source, (**f**) $^{14}NO_2^-$ as the N-source and $D_2O$ as the D source, (**g**) $^{15}NO_2^-$ as the N-source and $H_2O$ as the H source and (**h**) $^{15}NO_2^-$ as the N-source and $D_2O$ as the D source. **i** Schematic illustration of the cyclohexanone oxime generation pathway. **j** Yield and selectivity of different substrates. **k** Photograph of the circular flow electrolyzer. **l** Time-dependent cyclohexanone oxime yield at a current density of $-50\ mA\ cm^{-2}$ (the inset is a photograph of the produced cyclohexanone oxime).

$NO_2^-$ electroreduction without the addition of cyclohexanone. The concentration of $NH_2OH$ was determined by ion chromatography. As a result, no $NH_2OH$ was detected during and after the reaction. Theoretical calculations were further conducted to explain this phenomenon. As shown in Supplementary Figs. 18 and 19 and Supplementary Note 7, $NH_2OH^*$ desorption is an endothermic process, while its further reduction is an exothermic process. This indicates that the formed $NH_2OH^*$ is easier to further reduce to $NH_3$ rather than desorbing from the catalyst surface. This result explains why $NH_2OH$ was not detected

in the electrolyte. Therefore, cyclohexanone oxime is formed by condensation of $NH_2OH^*$ and cyclohexanone on the catalyst surface rather than in the electrolyte solution.

Thus, the reaction pathway is proposed based on the above discussion (Fig. 3i). The electroreduction of $NO_2^-$ first proceeds on the Cu-S surface solely in the following reaction pathway: $NO_2^- \rightarrow NO^* \rightarrow NHO^* \rightarrow NH_2O^* \rightarrow NH_2OH^*$(Ref. [28]). Then, the adsorbed cyclohexanone is rapidly attacked by nucleophilic $NH_2OH^*$ to yield cyclohexanone oxime (Path **I**). Meanwhile, the further

electroreduction of $NH_2OH^*$ to $NH_4^+$ also proceeds as a competing reaction. Delightedly, $NH_4^+$ can be converted to $NO_2^-$ by the electro-oxidation cycle[29] (Supplementary Fig. 20 and Supplementary Note 8) or collected as ammonium phosphate (a valuable fertilizer) (Path II).

## Universality and application evaluation of the electrosynthesis strategy

The universality of our method was evaluated. Other substrates of ketones and aldehydes, such as furfural, cyclopentanone, and cyclo-heptanone, are transformed to the corresponding oximes with high yields (72%–97%) and selectivity (>99%) (Fig. 3j, Supplementary Figs. 21–33 and Supplementary Notes 9–21). To evaluate its application potential, a circular flow electrolyzer was adopted to perform two-electrode tests using the Cu-S catalyst as the cathode and titanium mesh as the anode (Fig. 3k, Supplementary Fig. 34 and Supplementary Note 22). Constant current tests were performed at current values of 30, 40, 50, and 60 mA cm$^{-2}$ for 6 h of electrolysis (Fig. 3l and Supplementary Fig. 35). The optimal productivity of 40 mmol (4.52 g) cyclo-hexanone oxime with a production rate of 6.6 mmol h$^{-1}$ is obtained at a current value of −50 mA cm$^{-2}$. These results suggest promising applications of this electrocatalytic strategy in the production of various oximes with good substrate tolerance.

## Discussion

In conclusion, we report an electrochemical strategy to synthesize cyclohexanone oxime by utilizing $NH_2OH^*$ generated in situ by $NO_2^-$ electroreduction. This strategy avoids the use of $H_2O_2$, $H_2$, $SO_2$, high temperature, and noble metal catalysts that are required for the conventional approach. Up to 92% yield and 99% selectivity (in terms of C) of cyclohexanone oxime are obtained over a Cu-S cathode. The catalytic performance can be maintained well during 50 cycles of tests. The combined results of in situ ATR-SEIRAS, control experiments, and DFT calculations reveal that the reaction undergoes the processes of $NO_2^-$ RR to $NH_2OH^*$ and the condensation of $NH_2OH^*$ with cyclohexanone to cyclohexanone oxime. In addition, this method is suitable for synthesizing other oximes, highlighting the universality. Furthermore, the application potential of this strategy is elucidated by an amplified electrolysis experiment and TEA. This work opens a door for the mild, economical, and sustainable production of cyclohexanone oxime, which may be an alternative/complementary to the current industrial production of cyclohexanone oxime. Furthermore, our method may have wider applications in other industrial processes that require the use of $NH_2OH$.

## Methods

### Synthesis of self-supported Cu(OH)$_2$ nanowire arrays (NAs)

Cu(OH)$_2$ NAs were synthesized with slight modification according to a previous report[30]. Commercial copper foam (CF) was cut into a rectangular shape with a size of $1.0 \times 3.0$ cm$^2$. Then, the small pieces of CF were carefully washed with 3.0 M acid, acetone, and deionized water, respectively. Cu(OH)$_2$ NAs supported on CF were self-grown by simple oxidization of the Cu substrate in an alkaline environment. NaOH (3.0 g) and 0.68 g $(NH_4)_2S_2O_8$ were dissolved in 30 mL deionized (DI) water under vigorous stirring, and the precleaned Cu substrate was immersed in it. After 20 minutes, a blue hydroxide layer was observed on the surface of Cu. The Cu substrate covered with nanowires was removed from the solution, rinsed repeatedly with DI water, and then dried at room temperature.

### Synthesis of self-supported CuS nanowire arrays (CuS NAs)

CuS NAs were prepared by the reported hydrothermal sulfidation method[30]. Thiourea (0.22 g) was dissolved in 30 mL ethylene glycol under magnetic stirring at room temperature. Then, the solution was loaded into a 50 ml Teflon-lined autoclave, and a piece of freshly treated CF was also added into the autoclave, sealed, heated to 80 °C

and kept at this temperature for 60 min. After the reaction cooled naturally, the CF with products on it was removed, washed with water, and dried naturally.

### Synthesis of Cu-S nanowire sponges via in situ electroreduction of CuS NAs

Cu-S was synthesized with slight modification according to our previous report[23]. The electroreduction of CuS was conducted on an Ivium-n-Stat electrochemical workstation (Ivium Technologies B.V.) in a typical three-electrode system in 1.0 M KOH. A Hg/HgO with 1.0 M KOH as the inner reference electrolyte was used as the reference electrode. A carbon rod was used as the counter electrode. CuS NAs/CF was sealed in advance with epoxy to ensure an exposed area of 1.0 cm$^2$ and then used as the working electrode. Linear sweep voltammetry (LSV) was recorded in the voltage range −0.7 ~ −1.5 V vs. Hg/HgO at a scan rate of 5 mV s$^{-1}$ until the reductive peaks disappeared.

### Synthesis of Cu nanowire arrays (Cu NAs) via in situ electroreduction of Cu(OH)$_2$ NAs

Cu(OH)$_2$-derived Cu NAs were synthesized by a similar in situ electroreduction strategy to those of Cu-S NSs, where CuS NAs were replaced by Cu(OH)$_2$ NAs.

### Characterization

The in situ ATR-FTIR was performed on a Nicolet 6700 FTIR spectrometer with silicon as the prismatic window. $^1$H and $^{13}$C NMR were recorded on a Bruker AVANCE III 400 M NMR instrument. GC–MS was measured on an Agilent 5977B mass spectrometer with an Agilent Technologies 8860 GC system.

### Electrochemical measurements

The electrochemical measurement was carried out on an Ivium-n-Stat electrochemical workstation (Ivium Technologies B.V.) with an H-type cell (Supplementary Fig. 2). Meanwhile, the Cu or Cu-S, carbon rod, and Ag/AgCl electrode were adopted as the working electrode (the working area is 1.0 cm$^2$), the counter electrode, and the reference electrode, respectively. A Nafion 117 proton exchange membrane was applied to separate the anode and cathode compartments of the H-type cell. 20 mL of 0.5 M pH 5.8 PBS with 0.2 mmol cyclohexanone and 2 mmol NaNO$_2$ was used as the catholyte, while 20 mL of 0.5 M pH 5.8 PBS was used as the anolyte. For the linear sweep voltammetry (LSV) test, the constant potential was set as −0.3 V to −1.0 V (vs. Ag/AgCl) with a scan rate of 10 mV s$^{-1}$. For the constant potential electrolysis test, the potential range was set as −0.6 to −1.0 V (vs. Ag/AgCl), and the reaction time was 4000 s. The products were analysed and quantified by NMR spectroscopy, GC–MS and GC.

### Amplified electrolysis experiments

The amplified electrolysis experiments were conducted on a flow electrolytic cell (Fig. 3k, Supplementary Fig. 27). The electrolyzer is a two-electrode divided cell, and the anode and cathode are separated by a Nafion membrane. Cu-S catalyst and stainless-steel mesh were used as the cathode and anode, respectively. Before the reaction for cyclohexanone oxime electrosynthesis, 800 mL PBS (pH=5.8) was added to the anode and cathode reservoirs, and the cathodic CuS catalyst was electroreduced at a constant current of −30 mA cm$^{-2}$ for 20 min to obtain a Cu-S cathode. After that, 15 g cyclohexanone was first dissolved in 20 mL methanol and subsequently added to the cathode reservoir. The flow rate of the electrolyte was 100 L/h, and the current densities were 30, 40, 50, and 60 mA cm$^{-2}$. The products were analysed and quantified by GC–MS and GC.

### Product identification and quantification

The products in the electrolyte were identified by NMR spectroscopy and GC–MS. Cyclohexanone oxime production was quantified by GC

with cetane as the internal standard, and $NH_4^+$ was quantified by $^1H$ NMR with maleic acid as the internal standard. The amount of the analyte was calculated based on the area ratio of the analyte peak to that of the internal standard. For the identification and quantification of organic products, after electrolysis, after the reactions finished, the products were extracted by dichloromethane (DCM) and analysed by NMR spectroscopy and GC–MS. For the identification and quantification of $NH_4^+$ from $NO_2^-$ electroreduction, an extra 20 μL of 4.0 M $H_2SO_4$ was added to the as-prepared NMR sample to reach a pH value of ~3, and the concentration was calculated based on the area ratio of the $NH_4^+$ peak ($NH_4^+$, ~ 6.90 ppm, double peak) to that of maleic acid using calibration curves.

The yield was calculated by below equation:

$$Yield(\%) = \frac{mol\ of\ the\ formed\ product}{mol\ of\ the\ initial\ substrate} \times 100\% \quad (1)$$

The yield rate was calculated by below equation:

$$Yield\ rate\left(\frac{mmol}{h}\right) = \frac{mol\ of\ the\ formed\ product}{mol\ of\ the\ initial\ substrate \times t \times s} \times 100\% \quad (2)$$

where $t$ is the reaction time and $s$ is the geometric area of the electrode.

The conversion was calculated by below equation,

$$Conversion(\%) = \frac{mol\ of\ the\ consumed\ substrate}{mol\ of\ the\ initial\ substrate} \times 100\% \quad (3)$$

The Faradaic efficiency (FE) is the ratio of the number of electrons transferred for the formation of each product to the total amount of electricity passing through the circuit. The FE for the products was calculated using below equation:

$$FE(\%) = \frac{b \times n \times F}{Q} \times 100\% \quad (4)$$

where $F$ is the Faraday constant, $Q$ is the electric quantity, $n$ is the mole of generated products, and $b$ is the electron transfer number.

In this paper, error bars correspond to the standard deviation of three independent measurements.

## Electrochemical in situ ATR-FTIR spectra measurements

The in situ ATR-FTIR was performed on a Nicolet 6700 FTIR spectrometer equipped with an MCTA detector with silicon as the prismatic window. First, CuS ink (pure ethanol as a dispersant) was carefully dropped on the surface of the gold film, which was chemically deposited on the surface of the silicon prismatic before each experiment. Then, the deposited silicon prismatic served as the working electrode. The Pt foil and Ag/AgCl electrode containing saturated KCl solution were used as the counter and reference electrodes, respectively. The 0.5 M PBS $H_2O/D_2O$ solution (pH = 5.8) with cyclohexanone and $NaNO_2/Na^{15}NO_2$ was employed as the electrolyte. Spectra were recorded at −0.9 V vs. Ag/AgCl. The background spectrum of the catalyst electrode was acquired at an open-circuit voltage before each systemic measurement.

## Data availability

The data that support the plots within this paper are available from the corresponding author upon reasonable request. The source data underlying Figs. 2 and 3 are provided as a Source Data file. Source data are provided with this paper.

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

## Acknowledgements

We acknowledge the National Natural Science Foundation of China (grant no. 22271213 to B.Z.) and the National Postdoctoral Science Foundation of China (grant no. 2022M722357 to Y.W.) for financial support. We also appreciate the kind help from Ms. Yang Liu for ATR-FTIR measurements.

## Author contributions

B.Z. and Y.W. conceived the idea and designed the research. J.Z. and Y.W. conducted experiments and data analysis. Z.S. assisted in some experiments. C.W. conducted the calculations. T.L. and Y.W. performed the in situ ATR-FTIR measurements. B.H.Z. performed the techno-economic analysis. C.L. contributed to the discussion. Y.W. wrote the manuscript. B.Z. revised the paper with comments from all authors.

## Competing interests

The authors declare no competing interests.
