## [Peer Review File · Nature Communications]

Electrosynthesis of a nylon-6 precursor from cyclohexanone and nitrite under ambient conditionsREVIEWER COMMENTS

Reviewer #1 (Remarks to the Author):

The electrocatalytic synthesis of a range of oximes (primarily cyclohexanone oxime) is investigated, utilising a Cu-S-based catalyst under ambient conditions, with excellent selectivities towards the oxime reported, avoiding the need for commercial H₂O₂ that is utilised in the industrial approach.

While the formation of the oxime is a useful example for application, the conversion of nitrite to NH₂OH may be considered the actual reaction of interest and one that has been well studied within the literature. Indeed, the impressive rates of conversion reported can be considered a measure of the efficiency of oxime formation (ie of the reaction between ketone and NH₂OH), rather than related to the efficacy of hydroxylamine formation from nitrite. The authors report that the addition of ketone after the formation of hydroxylamine yields a similar result to when the ketone is present. I am concerned that the authors consider that 'These results demonstrate that cyclohexanone oxime production is an electrically driven process' (line 118), given that they observe a reaction between hydroxylamine and the ketone in the liquid phase, in the absence of an electrical current. However, they further report that 'To determine whether the condensation reaction occurs on the catalyst surface or in the solution phase, we added cyclohexanone to the electrolyte solution after electrolysis, and no cyclohexanone oxime product is formed' (lines 144-146). These additional experiments may indicate that the NH₂OH is retained on the catalyst surface, however, it is unclear if the reaction between ketone and NH₂OH occurs on the surface or if a current is required to desorb NH₂OH into the liquid phase. Additional experiments to support the hypothesis that the reaction is surface-mediated should be included and the clarity of the text improved in light of the seemingly contradictory results reported.

The reviewer queries how it is possible for no other organic by-products to be observed and for selectivity and conversion rates of 99% to be reported but only achieve a 92% yield of oxime? Additionally, Fig 2 d seemingly reports a selectivity of 100% in all cases, yet the yield of oxime does not match ketone conversion. Can the absence of by-products be truly confirmed by NMR analysis, what are the limits of detection and is there a potential for adsorption of these by-products? Notably in the thermal-catalytic approach, there is a potential for aldol condensation products and the water-mediated conversion of oxime to ketone. Additionally, there is a small unattributed signal in the 'after-electrolysis' H-NMR spectra of Fig 2a. Does this species account for the discrepancies in conversion and yield? Further experiments utilising cyclohexanone oxime (in the presence and absence of nitrite and ketone) should be included.

The authors claim that selectivities towards cyclohexanone oxime and rates of cyclohexanone conversion gain an economic advancement over current technologies. However, such techno-economic analysis for the existing route is not provided. The reviewer also highlights that the details associated with the technoeconomic analysis is not reported within the SI (as indicated by the text) but rather in the main manuscript. A thorough re-read of the text for further typos is necessary.

A significant excess of NaNO₂ is used. Indeed, the authors report that approximately 6 T of NaNO₂ is required to generate 1 T of oxime. Has the nitrite: ketone ratio been evaluated required and can the

authors comment on the selective utilisation of NaNO_2 .

The authors have reported the loss of S during the electrochemical reduction of CuS. To what extent is the S retained on the catalytic surface after extended use in the electrochemical production of cyclohexanone oxime? Additionally, do the authors observe the loss of Cu over 50 cycles of use? A comparison to the Cu-only catalyst would also be useful.

Minor comments:

It is unclear if Figs S.14-26 are NMR spectra of analytical standards or post-reaction solutions. If it is the former, this seems unnecessary, if the latter the reviewer is surprised that no ketone is observed.

The authors state that "Delightedly, NH_4^+ can be converted to NO_2^- by the electrooxidation cycle or collected as ammonium phosphate (a valuable fertilizer) (Path II)." Notably, ammonium oxidation was not investigated in this paper. Such claims should be supported with experimental data.

The use of the "hydrogenation" may be confusing as H_2 is not used in this work.

Some minor typos can be found in the captions of the supporting information.

Reviewer #2 (Remarks to the Author):

In this work, the authors report an electrocatalytic method to synthesize cyclohexanone oxime, the precursor of nylon-6, with excellent selectivity and yield, avoiding the use of H_2O_2 that is utilized in industrial approaches. A series of experimental evidence provides solid proof for supporting mechanistic insight. Scaling up experiments and TEA analysis suggest the potential of the proposed method. The method is applicable for synthesizing other oximes, suggesting the promising potential of the methodology. Given the importance of the target product, the significant synthetic advantages of the proposed strategy and the high quality of this manuscript, I believe the work is an important advance that is of interest to the industry and academia. Therefore, I would like to highly recommend the publication of this work in Nature Communications after minor revision.

1. Characterization like SEM and XPS of Cu-S catalyst before and after reaction should be provided in SI.
2. It is proposed to replace "Electrochemical transformation has emerged as an attractive strategy in synthetic chemistry" with "Electrochemistry has emerged as an attractive strategy in synthetic chemistry".
3. "Electrosynthesis of the nylon-6 precursor from cyclohexanone and nitrite under ambient conditions" should be a more attractive title.

4. The acquisition time of each scan in the infrared spectra should be indicated.
5. What is the solubility of cyclohexanone oxime in the aqueous phase? Is a co-solvent added?
6. The authors have demonstrated that other substrates of ketones and aldehydes, such as furfural, cyclopentanone, and cycloheptanone, can be transformed to the corresponding oximes. Thus, "this electrocatalytic strategy is suitable for the production of various oximes, highlight the methodology universality." should be added to the abstract.

A point-by-point response to the reviewers

To Reviewer 1:

Reviewer Letter:

The electrocatalytic synthesis of a range of oximes (primarily cyclohexanone oxime) is investigated, utilising a Cu-S-based catalyst under ambient conditions, with excellent selectivities towards the oxime reported, avoiding the need for commercial H_2O_2 that is utilised in the industrial approach.

Answer: We sincerely acknowledge the reviewer for reviewing our manuscript and making constructive suggestions. We address all the concerns and carefully revised the manuscript according to these professional comments from the reviewer, and the quality of the manuscript will surely be improved. To save the reviewers' valuable time, key revisions are displayed in a yellow background in the revised manuscript and supporting information (SI).

Comment 1. While the formation of the oxime is a useful example for application, the conversion of nitrite to NH_2OH may be considered the actual reaction of interest and one that has been well studied within the literature. Indeed, the impressive rates of conversion reported can be considered a measure of the efficiency of oxime formation (ie of the reaction between ketone and NH_2OH), rather than related to the efficacy of hydroxylamine formation from nitrite. The authors report that the addition of ketone after the formation of hydroxylamine yields a similar result to when the ketone is present. I am concerned that the authors consider that 'These results demonstrate that cyclohexanone oxime production is an electrically driven process' (line 118), given that they observe a reaction between hydroxylamine and the ketone in the liquid phase, in the absence of an electrical current. However, they further report that 'To determine whether the condensation reaction occurs on the catalyst surface or in the solution phase, we added cyclohexanone to the electrolyte solution after electrolysis, and no cyclohexanone oxime product is formed' (lines 144-146). These additional experiments may indicate that the NH_2OH is retained on the catalyst surface, however, it is unclear if the reaction between ketone and NH_2OH occurs on the surface or if a current is required to desorb NH_2OH into the liquid phase. Additional experiments to support the hypothesis that the reaction is surface-mediated should be included and the clarity of the text improved in light of the seemingly contradictory results reported.

Answer: We acknowledge the reviewer's comments. We have made vague expression in the mechanism section and caused misunderstanding. Now, we explain it as follows: the reaction undergoes the processes of NO_2^- reduction to hydroxylamine and the condensation of hydroxylamine with cyclohexanone to form cyclohexanone oxime. The reaction between ketone and hydroxylamine is a spontaneous process that requires no electricity. It seems that the reduction of nitrite to hydroxylamine is the actual reaction. However, when cyclohexanone was added to the electrolyte solution at the end of

NO_2^- reduction, no cyclohexanone oxime formation was detected. We speculate that the adsorbed hydroxylamine is difficult to desorb from the catalyst surface into the electrolyte solution to react with cyclohexanone. To prove this hypothesis, we performed an electrolysis experiment of NO_2^- electroreduction without the addition of cyclohexanone. The concentration of hydroxylamine was determined by ion chromatography. As a result, no detectable amount of hydroxylamine was observed during or after the reaction. It can exclude the possibility of the requirement of current to desorb hydroxylamine, as the reviewer noted. Theoretical calculations were further conducted to explain this phenomenon. As shown in **Figure R1**, hydroxylamine desorption is an endothermic process, while its hydrogenation is an exothermic process. This indicates that the formed hydroxylamine is easier to further reduce to NH_3 rather than desorbing from the catalyst surface. This result explains why hydroxylamine was not detected in the electrolyte. Therefore, cyclohexanone oxime is formed by condensation of adsorbed hydroxylamine and cyclohexanone on the catalyst surface rather than in the electrolyte solution. We have modified the expression in the mechanism section as follows:

“Control experiments reveal that when NH_2OH and cyclohexanone are mixed at room temperature, cyclohexanone oxime is immediately generated even without electricity, indicating that the condensation of NH_2OH and cyclohexanone is a spontaneous process (Entry 9 in Table 1). This inspired us to explore whether cyclohexanone oxime can be formed by adding cyclohexanone at the end of NO_2^- electroreduction. As a result, no cyclohexanone oxime was detected. We speculate that the adsorbed hydroxylamine is difficult to desorb from the catalyst surface into the electrolyte solution to react with cyclohexanone. To prove this hypothesis, we performed an electrolysis experiment of NO_2^- electroreduction without the addition of cyclohexanone. The concentration of NH_2OH was determined by ion chromatography. As a result, no NH_2OH was detected during or after the reaction (Supplementary Fig. 12). Theoretical calculations were further conducted to explain this phenomenon. As shown in Supplementary Figs. 13 and 14, NH_2OH^ desorption is an endothermic process, while its further reduction is an exothermic process. This indicates that the formed NH_2OH^* is easier to further reduce to NH_3 rather than desorbing from the catalyst surface. This result explains why NH_2OH was not detected in the electrolyte. Therefore, cyclohexanone oxime is formed by condensation of NH_2OH^* and cyclohexanone on the catalyst surface, rather than in the electrolyte solution.”*

Figure R1. (Supplementary Figure 17). Free energy diagram for cyclohexanone oxime generation over a Cu-S cathode.

Comment 2. The reviewer queries how it is possible for no other organic by-products to be observed and for selectivity and conversion rates of 99% to be reported but only achieve a 92% yield of oxime? Additionally, Fig 2d seemingly reports a selectivity of 100% in all cases, yet the yield of oxime does not match ketone conversion. Can the absence of by-products be truly confirmed by NMR analysis, what are the limits of detection and is there a potential for adsorption of these by-products? Notably in the thermal-catalytic approach, there is a potential for aldol condensation products and the water-mediated conversion of oxime to ketone. Additionally, there is a small unattributed signal in the ‘after-electrolysis’ H-NMR spectra of Fig 2a. Does this species account for the discrepancies in conversion and yield? Further experiments utilising cyclohexanone oxime (in the presence and absence of nitrite and ketone) should be included.

The authors claim that selectivities towards cyclohexanone oxime and rates of cyclohexanone conversion gain an economic advancement over current technologies. However, such techno-economic analysis for the existing route is not provided. The reviewer also highlights that the details associated with the techno-economic analysis is not reported within the SI (as indicated by the text) but rather in the main manuscript. A thorough re-read of the text for further typos is necessary.

A significant excess of NaNO_2 is used. Indeed, the authors report that approximately 6 T of NaNO_2 is required to generate 1 T of oxime. Has the nitrite: ketone ratio been evaluated required and can the authors comment on the selective utilisation of NaNO_2 .

Answer: Thank you for the reviewer’s wise comments. To verify whether there are byproducts with concentrations below the NMR detection limit, we used more sensitive GC–MS to detect the crudes after the reaction. No other byproducts were observed in the GC–MS spectrum (**Figure R2 and**

Supplementary Figure 8). Moreover, we extracted the reaction solution with ethyl acetate and carried out NMR analysis of the products. Highly pure NMR spectra were obtained without product separation and purification processes (**Figure R3 and Supplementary Figures 6, 7**). These results prove the high purity of the product. Furthermore, we performed electrolysis experiments using cyclohexanone oxime as the substrate in the presence and absence of nitrite and ketone, as the reviewer suggested. The products were analysed by GC–MS, and no amine or ketone products were detected, excluding the transformation of cyclohexanone oxime during the reaction (**Figures R4, R5**).

Based on the above experimental results, we speculate that the lower product yield than the substrate conversion is due to the **volatility** of cyclohexanone. In a simple test, 20 mL of electrolyte solution containing 0.2 mmol cyclohexanone was stirred for 4000 s (the actual reaction time) and 3 h at room temperature. As a result, 11% and 35% loss of cyclohexanone was determined by GC. The results explain the slightly lower product yield than the substrate conversion, while no other byproducts were detected. Thus, it is more accurate to use cyclohexanone oxime yield to measure the reaction process, and the cyclohexanone conversion data in Fig. 2d have been removed in the revised manuscript.

For techno-economic analysis (TEA), it is difficult to evaluate the cost of industrial processes, and the profit space resulting from TEA given by the current literature is usually with respect to the market price (*Nat Sustain* 4, 911–919 (2021); *Nat Catal* 5, 185–192 (2022)). In our work, the calculated cost of cyclohexanone oxime is 6328.04 \$/t, gaining a profit space of 3807.6 \$/t compared with the market price of 10115.6 \$/t (<http://www.100ppi.com/>). Detailed errors pointed out by the reviewer have been modified, and the whole manuscript has been checked.

In this work, 10-fold molar equivalents of NaNO_2 were used to realize the rapid synthesis of cyclohexanone oxime (0.2 mmol substrate, 92% oxime yield, 4000 s). When the molar ratios of NaNO_2 and cyclohexanone were reduced to 5:1, 2:1 and 1:1, the yields of cyclohexanone oxime decreased to 73%, 52% and 35%, respectively, within the same reaction time (**Figure R6 and Supplementary Figure 15**). Based on the mechanism analysis of the reaction, the utilization of nitrite can be improved from the aspects of catalyst and reactor design. Hydroxylamine is the key species for cyclohexanone oxime formation, so developing electrocatalysts that can selectively produce hydroxylamine products during nitrite electroreduction is supposed to effectively increase nitrite utilization. However, there are few reports of hydroxylamine as the main product in nitrate/nitrite electroreduction reactions. This is because most catalysts have strong adsorption of hydroxylamine intermediates and easily reduce them to ammonia, which makes it difficult for them to desorb from the catalyst surface to the solution to form hydroxylamine products. Thus, combining theoretical calculations and experiments to modify catalytic materials to reduce their adsorption of hydroxylamine intermediates and improve their further reduction energy barrier is the key to improving nitrite utilization. Furthermore, for catalysts that have

difficulty desorbing hydroxylamine, the condensation of hydroxylamine and ketone occurs on the catalyst surface. Therefore, matching nitrite reduction and ketone mass transfer rates is also significant for improving the utilization rate of nitrite. In general, catalysts can adsorb nitrite more easily than ketone, so it is necessary to enhance the mass transfer of ketone. On the one hand, the hydrophilic/oleophilic properties of the catalyst can be regulated to balance the adsorption of nitrite and ketone. On the other hand, a flow electrocatalytic reaction device can be constructed to match the mass transfer by adjusting the flow rate of the reaction liquids.

Figure R2 (Supplementary Figure 8). GC-MS spectra of the reaction product obtained under standard conditions.

Figure R3 (Supplementary Figure 6, 7). ¹H and ¹³C NMR spectra of cyclohexanone oxime product.

Figure R4. GC-MS spectra of the reaction product obtained under the conditions of using cyclohexanone oxime as the substrate in the presence of nitrite and ketone.

Figure R5. GC–MS spectra of the reaction product obtained under the conditions of using cyclohexanone oxime as the substrate in the absence of nitrite and ketone.

Figure R6 (Supplementary Figure 15). Cyclohexanone oxime yield and FE under the conditions of different molar ratios of cyclohexanone and NO₂⁻.

Comment 3. The authors have reported the loss of S during the electrochemical reduction of CuS. To what extent is the S retained on the catalytic surface after extended use in the electrochemical production of cyclohexanone oxime? Additionally, do the authors observe the loss of Cu over 50 cycles of use? A comparison to the Cu-only catalyst would also be useful.

Answer: Thank you very much for the reviewer's comments. The S content of the Cu-S catalysts after 50 cycles of reaction is approximately 3.8% (Atom%), as determined by X-ray photoelectron spectroscopy. Regarding the reviewer's concern about the loss of Cu, we performed an inductively coupled plasma-mass spectrometry (ICP-MS) test of the solution after the reaction. No detectable amount of Cu was observed, excluding the loss of Cu during the reaction.

Comment 4. It is unclear if Figs S.14-26 are NMR spectra of analytical standards or post-reaction solutions. If it is the former, this seems unnecessary, if the latter the reviewer is surprised that no ketone is observed.

Answer: Thank you very much for the reviewer's comments. Figs. S.14-26 show the NMR spectra of the oxime products purified from postreaction solutions. Because of the extraction and purification processes, no ketones are observed in the spectra.

Comment 5. The authors state that "Delightedly, NH_4^+ can be converted to NO_2^- by the electrooxidation cycle or collected as ammonium phosphate (a valuable fertilizer) (Path II)." Notably, ammonium oxidation was not investigated in this paper. Such claims should be supported with experimental data.

Answer: Thank you very much for the reviewer's comments. The electrooxidation of NH_4^+ to NO_2^- has been experimentally demonstrated, and the results are shown in **Figure R7 (Supplementary Figure 20)**. The electrooxidation of NH_4^+ to NO_2^- is conducted in a two-compartment three-electrode system. 0.1 M KOH solution containing 10 mM NH_4Cl was used as the electrolyte. $\text{Cu}(\text{OH})_2$ supported on Cu foam, denoted $\text{Cu}(\text{OH})_2/\text{Cu}$ NF, was used as the working electrode. The electrolytic reaction proceeded at potentials of 1.5 to 1.7 V vs. RHE. The produced NO_2^- was quantified by the UV-vis absorption spectrum after passing a charge of 260 C. Impressively, 42% FE and 95% yield of NO_2^- were obtained at the optimal potential of 1.6 V vs. RHE, rationalizing the recycling idea of the N source.

Figure R7 (Supplementary Figure 20) (a) The calibration curves of $\text{NH}_3\text{-N}$ based on the absorbance of different ion concentrations¹ and (b) the FEs and yields of NO_2^- for NH_4^+ electrooxidation at different potentials over a $\text{Cu}(\text{OH})_2$ electrode.

Comment 6. The use of the "hydrogenation" may be confusing as H_2 is not used in this work.

Answer: Thank you very much for your kind suggestion. We have replaced "hydrogenation" with

“reduction” in the manuscript.

Comment 7. Some minor typos can be found in the captions of the supporting information.

Answer: Thank you very much for your comment. We have checked the whole manuscript carefully and revised these typos.

To Reviewer 2:

Reviewer Letter: In this work, the authors report an electrocatalytic method to synthesize cyclohexanone oxime, the precursor of nylon-6, with excellent selectivity and yield, avoiding the use of H_2O_2 that is utilized in industrial approaches. A series of experimental evidence provides solid proof for supporting mechanistic insight. Scaling up experiments and TEA analysis suggest the potential of the proposed method. The method is applicable for synthesizing other oximes, suggesting the promising potential of the methodology. Given the importance of the target product, the significant synthetic advantages of the proposed strategy and the high quality of this manuscript, I believe the work is an important advance that is of interest to the industry and academia. Therefore, I would like to highly recommend the publication of this work in *Nature Communications* after minor revision.

Answer: We are very grateful to the reviewers for their positive comments on the significance of our work. According to the reviewers' comments, we have addressed all the issues. To save reviewers valuable time, key revisions are displayed in a yellow background in the revised manuscript and supporting information (SI).

Comment 1. Characterization like SEM and XPS of Cu-S catalyst before and after reaction should be provided in SI.

Answer: We appreciate the reviewer's kind comments. SEM and XPS of Cu-S catalyst before and after reaction should be provided in SI.

Figure R1 (Supplementary Figure 2) SEM image of Cu-S catalyst before reaction.

Figure R2 (Supplementary Figure 3) Cu 2p and S 2p spectra of the Cu-S catalyst before the reaction.

The peaks at 932.2 and 953 eV were assigned to the $\text{Cu}^+ 2p_{3/2}$ and $\text{Cu}^+ 2p_{1/2}$ of Cu_2O , whereas the small peaks at 933.5 eV and 954.1 with the satellite peak at 942 eV correspond to the Cu^{2+} of CuO because of its oxidation in air.

Figure R3 (Supplementary Figure 16) SEM image of Cu-S catalyst after stability test.

Figure R4 (Supplementary Figure 17) Cu 2p and S 2p spectra of the Cu-S catalyst after the stability test.

Comment 2. It is proposed to replace “Electrochemical transformation has emerged as an attractive strategy in synthetic chemistry” with “Electrochemistry has emerged as an attractive strategy in synthetic chemistry”.

Answer: We are grateful for the reviewer’s kind suggestion. We have modified the expression as “*Electrochemistry has emerged as an attractive strategy in synthetic chemistry*”.

Comment 3. “Electrosynthesis of the nylon-6 precursor from cyclohexanone and nitrite under ambient conditions” should be a more attractive title.

Answer: We are grateful for the reviewer’s kind suggestion. We have modified the title to “Electrosynthesis of the nylon-6 precursor from cyclohexanone and nitrite under ambient conditions”.

Comment 4. The acquisition time of each scan in the infrared spectra should be indicated.

Answer: We are grateful for the reviewer’s comment. The acquisition time of each scan in the infrared spectra has been added.

Comment 5. What is the solubility of cyclohexanone oxime in the aqueous phase? Is a co-solvent added?

Answer: Thank you very much for your comment. The solubility of cyclohexanone oxime is <0.1 g/100 mL at 20 °C. 1,4 dioxane was used as the cosolvent in the amplified electrolysis experiment, and no cosolvent was added in other experiments. The experimental details are shown in the SI.

Comment 6. The authors have demonstrated that other substrates of ketones and aldehydes, such as furfural, cyclopentanone, and cycloheptanone, can be transformed to the corresponding oximes. Thus, “this electrocatalytic strategy is suitable for the production of various oximes, highlight the methodology university.” should be added to the abstract.

Answer: Thank you very much for your comment. We have added “*this electrocatalytic strategy is suitable for the production of various oximes, highlighting the methodology university.*” to the abstract.

We acknowledge all the kind comments and wise suggestions from the two reviewers. We are sure that the quality of this work will be greatly improved according to these helpful comments and wise suggestions.

REVIEWERS' COMMENTS

Reviewer #1 (Remarks to the Author):

I thank the authors for thoroughly addressing the concerns raised in my initial review of this manuscript. The additional data and corresponding discussion included in this resubmitted work has certainly strengthened the quality of the work, which was already very high. I am supportive of the acceptance of the resubmitted manuscript in Nature Communications.

Reviewer #2 (Remarks to the Author):

Authors have addressed all concerns, the quality of paper has been improved. its publication is recommended.

A point-by-point response to the reviewers

To Reviewer 1:

Reviewer Letter:

I thank the authors for thoroughly addressing the concerns raised in my initial review of this manuscript. The additional data and corresponding discussion included in this resubmitted work has certainly strengthened the quality of the work, which was already very high. I am supportive of the acceptance of the resubmitted manuscript in Nature Communications.

Answer: We highly appreciate the reviewer for his/her positive comments on our revised manuscript. We are sure that the quality of this work has been greatly improved according to these nice comments and suggestions. Thanks very much.

To Reviewer 2:

Reviewer Letter: Authors have addressed all concerns, the quality of paper has been improved. its publication is recommended.

Answer: We highly appreciate the reviewer for his/her positive comments on our revised manuscript. We are sure that the quality of this work has been greatly improved according to these nice comments and suggestions. Thanks very much.